# Multivariate Distributionally Robust Convex Regression under Absolute Error Loss

**Jose Blanchet**
Stanford MS&E
jose.blanchet@stanford.edu

**Peter W. Glynn**
Stanford MS&E
glynn@stanford.edu

**Jun Yan**
Stanford Statistics
junyan65@stanford.edu

**Zhengqing Zhou**
Stanford Mathematics
zqzhou@stanford.edu

## Abstract

This paper proposes a novel non-parametric multidimensional convex regression estimator which is designed to be robust to adversarial perturbations in the empirical measure. We minimize over convex functions the maximum (over Wasserstein perturbations of the empirical measure) of the absolute regression errors. The inner maximization is solved in closed form resulting in a regularization penalty involves the norm of the gradient. We show consistency of our estimator and a rate of convergence of order $\widetilde{O}\left(n^{-1/d}\right)$, matching the bounds of alternative estimators based on square-loss minimization. Contrary to all of the existing results, our convergence rates hold without imposing compactness on the underlying domain and with no a priori bounds on the underlying convex function or its gradient norm.

## 1 Introduction

Convex regression estimation arises in a wide range of learning applications, for example, when fitting demand functions, production curves or utility functions, see [14, 22, 23]. Economic theory often dictates that demand functions are concave, [2]. In financial engineering, stock option prices often exhibit convexity restrictions [1]. This paper introduces a novel convex regression estimator which, by design, enjoys enhanced robustness properties. This estimator requires no a priori uniform bounds on the underlying convex function or its Lipschitz constant, nor does our estimator require that the domain of the convex function be compact, in contrast to existing convex function estimators that have known convergence rate guarantees. Furthermore, our numerical experiments show that our estimator exhibits good empirical performance, in comparison with existing estimators, and is a promising alternative to existing methods.

Let $X$ be a $d$-dimensional random vector and let $Y$ be a scalar random variable. Given a sample $(X_1, Y_1), \cdots, (X_n, Y_n)$ of i.i.d. copies of $(X, Y)$, we adopt the convex regression model

$$Y_i = f_*(X_i) + \mathcal{E}_i, \tag{1}$$

where $f_* : \mathbb{R}^d \to \mathbb{R}$ is a (unknown) convex function and $\mathcal{E}_i$ is a zero-median random variable independent of $X_i$, satisfying mild regularity conditions indicated in the sequel. Unlike the existing literature on convex regression (or, more generally, shape-based regression), we base our estimation methodology not on minimizing the squared error loss, but on minimizing mean absolute error loss. We adopt this viewpoint as a means of reducing the sensitivity of our regression estimator to outliers in the data.

We further wish to regularize our estimator. One vehicle towards accomplishing this goal in a principled fashion is to consider a distributionally robust formulation in which we robustify over a Wasserstein ball around the data, using a diameter that is driven by consistency and convergence rate considerations. When we do this, we arrive at a computationally tractable formulation of the problem that can be solved as a linear program. This is to be contrasted against the quadratic program that arises when minimizing squared error loss. Furthermore, the form of regularization that appears in this problem involves a novel gradient-based penalization term, to be described in more detail later in this Introduction.

In order to introduce our Wasserstein-based distributionally robust optimization formulation, we first recall how the Wasserstein distance is defined.

First, let $\mathcal{P}(\mathbb{R}^m \times \mathbb{R}^m)$ be the space of Borel probability measures defined on $\mathbb{R}^m \times \mathbb{R}^m$. Let $\Pi(\mu, \nu)$ be the subspace of $\mathcal{P}(\mathbb{R}^m \times \mathbb{R}^m)$ with fixed marginals given by $\mu$ and $v$, respectively. That is, if $U \in \mathbb{R}^m, V \in \mathbb{R}^m$ are random vectors with joint distribution $\pi \in \mathcal{P}(\mathbb{R}^m \times \mathbb{R}^m)$, then $\pi \in \Pi(\mu, \nu)$, if the marginal distribution of $U$, $\pi_U$, equals $\mu$ and the marginal distribution of $V$, $\pi_V$, equals $\nu$. The Wasserstein distance between $\mu$ and $\nu$ is given by

$$D(\mu, \nu) := \inf \left\{ \mathbb{E}_\pi \left[ c\left(U, V\right) \right] : \pi \in \mathcal{P}(\mathbb{R}^m \times \mathbb{R}^m), \pi_U = \mu, \pi_V = \nu \right\},$$

where $c : \mathbb{R}^m \times \mathbb{R}^m \to [0, \infty]$ is a metric. In our setting, we have $m = d + 1$, and we will choose as our metric

$$c\left((x, y), (x', y')\right) = \|x - x'\|_1 \mathbb{1}\left(y = y'\right) + \infty \mathbb{1}\left(y \neq y'\right). \tag{2}$$

We take the view here that distributional uncertainty is incorporated only in terms of the predictors and not the responses, since the responses already include a measurement error (in the term $\mathcal{E}$). This type of cost function has been used in the literature, [6], to exactly recover regularized estimators such as sqrt-Lasso, among others. It is possible to add distributional uncertainty in the response. The methods that we propose allow for adding distributional uncertainty in the response with only a small variation in the form of the estimator and without any change in the learning rates or the assumptions that we impose. Since the challenge here arises from the multidimensional aspect of the predictor variable, we decided to mostly impose the distributional robustness on the predictors.

Now, consider a loss function $l(y, z) : \mathbb{R} \times \mathbb{R} \to \mathbb{R}$, which is assumed to be convex and uniformly Lipschitz. Our distributionally robust convex regression (DRCR) formulation takes the form,

$$\inf_{f \in \mathcal{F}} \sup_{P \in \mathcal{P}(\mathbb{R}^{d+1}) : D(P, P_n) \leq \delta} \mathbb{E}_P \left[ l(Y, f(X)) \right], \tag{3}$$

where $\mathcal{F}$ represents the class of convex and Lipschitz functions (formally defined in Section 2.3), the parameter $\delta := \delta_n > 0$ is the uncertainty radius. This radius will be judiciously chosen as a function of $n$ to obtain consistency and suitable rates of convergence. The notation $P_n$ encodes the empirical distribution of the observations $(X_1, Y_1), \cdots, (X_n, Y_n)$, namely,

$$P_n(dx, dy) := \frac{1}{n} \sum_{i=1}^n \delta_{\{(X_i, Y_i)\}}(dx, dy).$$

Distributionally robust optimization formulations such as (3) have been used in a wide range of settings in the operations research literature and these formulations have become increasingly popular in machine learning and statistics.

**Our main contributions in this paper are as follows.**

i) We provide a tractable formulation of (3), in particular, we will show that

$$\inf_{f \in \mathcal{F}} \sup_{P \in \mathcal{P}(\mathbb{R}^{d+1}) : D(P, P_n) \leq \delta} \mathbb{E}_P \left[ l(Y, f(X)) \right] = \inf_{f \in \mathcal{F}} \left\{ \delta L \|\nabla f\|_\infty + \mathbb{E}_{P_n} l(Y, f(X)) \right\}, \tag{4}$$

where $\|\nabla f\|_\infty$ is the largest $l_\infty$-norm of all subgradients of $f(x)$ for all $x$, and similarly, $L := \sup_{(y,z) \in \mathbb{R} \times \mathbb{R}} |\nabla_z l(y, z)|$ (see Theorem 1). Note the penalty term is expressed in terms of the norm of the gradient of the estimator. The appearence of the $l_\infty$-norm is intimately connected to the choice of the $l_1$ cost function given in (2).

ii) Assuming that $l(y, f(x)) = |y - f(x)|$, we provide statistical guarantees for the rate of convergence of the estimators obtained in (4), improving upon the results obtained using a quadratic loss. In particular, we show that if $\|X\|_\infty^\gamma$ has a finite moment generating function in a neighborhood of the origin for some $\gamma > 0$ and if $\delta_n$ is chosen to be $\widetilde{O}\left(n^{-2/d}\right)$, then, under suitable regularity conditions on the residuals (see Theorem 2),

$$\widehat{f}_{n,\delta_n} = f^* + \widetilde{O}\left(n^{-1/d}\right),$$

in a suitable sense, where $\widehat{f}_{n,\delta_n} \in \arg\inf_{f \in \mathcal{F}} \{\delta_n L \|\nabla f\|_\infty + \mathbb{E}_{P_n} l(Y, f(X))\}$ and the notation $\widetilde{O}\left(n^{-1/d}\right)$ ignores poly-log factors in $n$. In contrast to the current results in the literature, our rate of convergence does not require $X$ to have compact support, nor do we need to build an apriori bound on the size of the gradient of $f$ into our estimator in order to obtain convergence rate result.

Our contributions have several significant features. First, it is not difficult to see that choosing the absolute error loss $l(y, f(x)) = |y - f(x)|$ makes (4) equivalent to a linear programming problem. In fact, since $P_n$ is finitely supported, the problem becomes a finite dimensional linear programming problem. Hence, this problem is, in principle, easier to solve than the standard quadratic problem that arises in typical non-parametric convex regression formulations, which arise when minimizing the squared error loss.

Second, our estimator is naturally endowed with desirable out-of-sample features due to the presence of the inner maximization, which explores the impact on the loss function due to statistical variations in the data. This interpretation follows from the left hand side of (4). The right hand side of (4), on the other hand, shows a direct connection to regularization in terms of the norm of the gradient of $f$, and the resulting norm is the dual transportation cost. This regularization term, as we shall see, allows us to construct an estimator that are free of a priori bounds imposed on the size of the gradient of $f$, which typically are required in order to obtain statistical guarantees. We now provide a literature review in the scientific areas touched by our contribution, namely, convex regression estimation and distributionally robust optimization.

## 1.1  Related Literature

In the context of convex regression, the overwhelming majority of the literature focuses on empirical least-squares estimators (leading to a quadratic programming formulation of the same size as the linear programming formulation that we offer). In one dimension, the work of [11] proves the consistency of the least squares estimator, and provides a rate of convergence of order $O(n^{-2/5})$ and an asymptotic distribution for this estimator; a matching upper and lower bounds for the min-max risk (in terms of quadratic loss) was obtained in [12], also with the same rate of order $O(n^{-2/5})$ up to a logarithmic factor. The first consistency results in higher dimensional problems were obtained in [16, 19]. Associated rates of convergence have only been derived recently, in [3, 13, 15], all of which assume that the predictor takes values on a compact set. It is shown in these papers that a phase transition occurs at $d = 4$. When $d \leq 4$, the least squares estimator achieves the convergence rate of $n^{-2/(d+4)}$, which matches the optimal convergence rate in the non-parametric setting (when $f_*$ is a twice continuously differentiable and the data is restricted to lie on a compact set). However, when $d > 4$, the convergence rate of the least squares estimator deteriorates to $O(n^{-1/d})$. Moreover, the results in [15] and [3] require apriori knowledge on $\|\nabla f_*\|_\infty$ in the construction of their estimator, while [13] requires knowledge of $\|f_*\|_\infty$. The work of [13] shows that under additional smoothness assumptions, the optimal min-max risk is of order $n^{-2/(d+4)}$, although, interestingly, no explicit estimator was given to recover such a rate in dimensions larger than four.

In connection to optimization, our formulation connects to an area which has been active in operations research for many years, namely, robust and distributionally robust optimization [5]. Distributionally robust optimization (DRO) problems informed by optimal transport costs, as in this paper's formulation, have become popular in recent years not only in operations research but also in the machine learning community. The work of [20] is the first one to show a connection to regularized estimators, in the context of logistic regression. The paper [6] provides an exact recovery of sqrt-Lasso and support vector machines. The work in [6] uses the DRO formulation to define a statistical criterion to optimally choose the uncertainty size $\delta$. This criterion, when applied to linear regression problems, recovers the scalings both in dimension and sample size obtained in the high-dimensional statistics

literature (see, for example, [4]). Applications in training of deep neural networks are given in [21], and additional representations of other estimators are given in [8, 10, 18], among others. A key step involved in obtaining these representations involves a duality result, which is given in [7].

## 1.2 Organization

The rest of this paper is organized as follows. In Section 2.1, we state and prove a strong duality result for the DRCR formulation in (6). Section 2.2 provides an explicit construction of the DRCR estimator, and in Section 2.3, we show that the convergence rate of this estimator is at most $\widetilde{O}(n^{-1/d})$. Finally we run a simulation study showing that the DRCR estimator can outperform the standard LSE or kernel based estimator. The proof of Theorem 2, as well as the main lemmas, is deferred to the supplementary materials.

## 2 Main Results

We first discuss our main result corresponding to the first contribution stated in the Introduction. We later turn to the second contribution. In order to state the strong duality result, we introduce some notations as follows. Let $x = (x_1, \cdots, x_d)$, denoted by $\partial f(x)$ the subdifferential of $f$ at $x$, and we define $\partial_{x_i} f(x)$ to be the partial subdifferential of $f$ at $x$ with respect to $x_i$. we define $\|\nabla f\|_\infty := \sup_{x \in \mathbb{R}^d} \max \{\|g\|_\infty : g \in \partial f(x)\}$, and $|\nabla_{x_i} f(x)| := \max \{|g| : g \in \partial_{x_i} f(x)\}$. Finally, let $\nabla f(x)$ denotes one of the solutions in $\arg \max \{\|g\|_\infty : g \in \partial f(x)\}$.

## 2.1 Dual formulation of DRCR

In this section, we establish the strong duality result for the DRCR problem (3), which plays an important role in the construction of our estimator and the analysis of rate of convergence.

**Theorem 1** (Strong Duality). *Suppose $l(y, z) : \mathbb{R} \times \mathbb{R} \to \mathbb{R}$ is a convex and Lipschitz function, such that $l(y, z) = l(-y, -z)$. Define*

$$L := \sup_{(y,z) \in \mathbb{R} \times \mathbb{R}} |\nabla_z l(y, z)|.$$

*Then, for any $\delta \geq 0$,*

$$\inf_{f \in \mathcal{F}} \sup_{P \in \mathcal{P}(\mathbb{R}^{d+1}): D(P, P_n) \leq \delta} \mathbb{E}_P \left[ l(Y, f(X)) \right] = \inf_{f \in \mathcal{F}} \left\{ \delta L \|\nabla f\|_\infty + \frac{1}{n} \sum_{i=1}^{n} l(Y_i, f(X_i)) \right\}.$$

By the above theorem, we see that the DRCR (3) problem is essentially equivalent to a regularized empirical loss, where the supremum norm of $\nabla f$ is penalized.

*Proof of Theorem 1.* To begin, we invoke the following lemma

**Lemma 1** ([7]). *Given any probability distribution $\mu \in \mathcal{P}(\mathbb{R}^d)$, for any upper semi-continuous function $f \in L_1(d\mu)$ and any cost function $c$, the following strong duality holds:*

$$\sup_{\nu \in \mathbb{P}(\mathbb{R}^d): D(\mu, \nu) \leq \delta} \mathbb{E}_\nu f(X) = \inf_{\lambda \geq 0} \left\{ \lambda \delta + \mathbb{E}_\mu \left[ \sup_{y \in \mathbb{R}^d} \{ f(y) - \lambda c(X, y) \} \right] \right\}.$$

As a direct consequence of Lemma 1, we have for any $f \in \mathcal{F}$ that

$$\sup_{\mathcal{P} \in \mathbb{R}^{d+1}: D(P, P_n) \leq \delta} \mathbb{E}_P \left[ l(Y, f(X)) \right]$$

$$= \inf_{\lambda \geq 0} \left\{ \lambda \delta + \mathbb{E}_{P_n} \left[ \sup_{(x,y) \in \mathbb{R}^d \times \mathbb{R}} \{ l(y, f(x)) - \lambda c\left( (X, Y), (x, y) \right) \} \right] \right\}$$

$$= \inf_{\lambda \geq 0} \left\{ \lambda \delta + \frac{1}{n} \sum_{i=1}^{n} \sup_{x \in \mathbb{R}^d} \{ l(Y_i, f(x)) - \lambda \|x - X_i\|_1 \} \right\}. \tag{5}$$

For simplicity, let $\nabla_i f(x)$ denotes the $i$th coordinate of $\nabla f(x)$, $(1 \le i \le d)$. Suppose $\lambda < L\|\nabla f\|_\infty$, then there exists $y_0 \in \mathbb{R}$, $z_0 \in \mathbb{R}$, $x_0 \in \mathbb{R}^d$ and $i_0 \in \{1, \ldots, d\}$, such that $\lambda < |\nabla_z l(y_0, z_0)| \cdot |\nabla_{i_0} f(x_0)|$. Without lost of generality, we may assume that $\nabla_z l(y_0, z_0) \nabla_{i_0} f(x_0) > 0$. Otherwise, we consider $(-y_0, -z_0)$. We may consider the case that both $\nabla_z l(y_0, z_0), \nabla_{i_0} f(x_0) > 0$, since the case in which both of them are negative is similar. Let $\{e_i\}_{i=1}^d$ be the canonical basis of $\mathbb{R}^d$, if $x_t := x_0 + t \cdot e_{i_0} \in \mathbb{R}^d$, then $f(x_t)$ is a convex function of $t$. Moreover, under the above assumptions, we have $f(x_t) \to +\infty$ as $t \to +\infty$. Hence, together with the convexity of $l$, for $t > 0$ sufficiently large,

$$
\begin{aligned}
& l(Y_i. f(x_t)) - \lambda \|x_t - X_i\|_1 \\
\ge\ & l(y_0, f(x_t)) - \lambda \|x_t - x_0\|_1 - L_0 |y_0 - Y_i| - \lambda \|x_0 - X_i\| \\
\ge\ & l(y_0, z_0) + \nabla_z l(y_0, z_0) \cdot (f(x_t) - z_0) - \lambda t - L_0 |y_0 - Y_i| - \lambda \|x_0 - X_i\| \\
\ge\ & (\nabla_z l(y_0, z_0) \nabla_{i_0} f(x_0) - \lambda) t + \nabla_z l(y_0, z_0) \cdot (f(x_0) - z_0) + l(y_0, z_0) - L_0 |y_0 - Y_i| \\
& - \lambda \|x_0 - X_i\|,
\end{aligned}
$$

where $L_0 := \sup_{(y,z) \in \mathbb{R} \times \mathbb{R}} |\nabla_y l(y, z)| < \infty$. By taking the supremum over $t$, we have

$$
\sup_{x \in \mathbb{R}^d} \{l(Y_i, f(x)) - \lambda \|x - X_i\|_1\} = \infty.
$$

On the other hand, if $\lambda \ge L\|\nabla f\|_\infty$, we have for any $x \in \mathbb{R}^d$ that

$$
l(Y_i, f(x)) - l(Y_i, f(X_i)) \le L\|\nabla f\|_\infty \|x - X_i\|_1 \le \lambda \|x - X_i\|_1,
$$

where the equality holds if $x = X_i$. Hence

$$
\sup_{x \in \mathbb{R}^d} \{l(Y_i, f(x)) - \lambda \|x - X_i\|_1\} = l(Y_i, f(X_i)).
$$

Now, we can rewrite the equation (5) as

$$
\begin{aligned}
\sup_{\nu \in \mathbb{P}(\mathbb{R}^d): D(\mu, \nu) \le \delta} \mathbb{E}_\nu f(X) = \ & \inf_{\lambda \ge L\|\nabla f\|_\infty} \left\{ \lambda \delta + \frac{1}{n} \sum_{i=1}^n l(Y_i, f(X_i)) \right\} \\
= \ & \delta L \|\nabla f\|_\infty + \frac{1}{n} \sum_{i=1}^n l(Y_i, f(X_i)).
\end{aligned}
$$

$\square$

## 2.2 Construction of the DRCR Estimator

To construct the DRCR estimator, we focus now on the absolute error loss $l(y, f(x)) = |y - f(x)|$. Consider the following class of convex and Lipschitz functions:

$$
\mathcal{F}_n := \{f : f \text{ is convex}, \|\nabla f\|_\infty \le \log n\}.
$$

It can be checked directly that the loss function $l$ satisfies the requirements in Theorem 1 with the constant $L = 1$, so, we can rewrite the DRCR problem (3) as follows:

$$
\inf_{f \in \mathcal{F}_n} \left\{ \delta \|\nabla f\|_\infty + \frac{1}{n} \sum_{i=1}^n l(Y_i, f(X_i)) \right\}. \tag{6}
$$

Now we construct an estimator $\widehat{f}_{n,\delta}$ that solve the problem (6). Consider the following finite dimensional linear programming (LP)

$$
\begin{aligned}
\min_{g_i, \xi_i} \quad & \frac{1}{n} \sum_{i=1}^n l(Y_i, g_i) + \delta \max_{1 \le i \le n} \|\xi_i\|_\infty. \\
\text{s.t.} \quad & g_j \ge g_i + \langle \xi_i, X_j - X_i \rangle, \quad 1 \le i, j \le n. \\
& |\xi_i^k| \le \log n, \text{ where } \xi_i = (\xi_i^1, \cdots, \xi_i^d), 1 \le i \le n.
\end{aligned} \tag{7}
$$

Let $(\widehat{g}_1, \widehat{\xi}_1), \cdots, (\widehat{g}_n, \widehat{\xi}_n)$ be any solution of problem (7). Then, we can define the DRCR estimator by

$$
\widehat{f}_{n,\delta}(x) := \max_{1 \le i \le n} \left( \widehat{g}_i + \langle \widehat{\xi}_i, x - X_i \rangle \right), \tag{8}
$$

where $\langle \cdot, \cdot \rangle$ is the standard inner product. Next, we show that $\widehat{f}_{n,\delta}$ also solves the problem (6). In fact, $\widehat{f}_{n,\delta}$ is a solution to the problem

$$\inf_{f \in \mathcal{F}_n} \left\{ \delta \sup_{1 \leq i \leq n} \|\nabla f(X_i)\|_\infty + \frac{1}{n} \sum_{i=1}^n l(Y_i, f(X_i)) \right\},$$

where the objective value certainly serves as a lower bound for that of (6). Moreover, observe that $\|\nabla \widehat{f}_{n,\delta}\|_\infty = \max_{1 \leq i \leq n} \|\widehat{\xi}_i\|_\infty = \sup_{1 \leq i \leq n} \|\nabla f(X_i)\|_\infty$, hence $\widehat{f}_{n,\delta}$ is also a solution of (6).

## 2.3 Rate of Convergence

In order to state our rate of convergence result, corresponding the second contribution stated in the Introduction, we need to impose some assumptions and state some definitions.

Let $\mathcal{P}(\mathbb{R}^n)$ denote the set of all probability measures supported on $\mathbb{R}^n$. Given a metric space $(\mathcal{X}, \rho)$ and any subset $\mathcal{G} \subset \mathcal{X}$, the $\varepsilon$-covering number $M(\mathcal{G}, \varepsilon; \rho)$ is defined as the smallest number of balls with radius $\varepsilon$ whose union contains $\mathcal{G}$, and let $A_\varepsilon$ denotes any corresponding $\varepsilon$-covering set. We say a random variable $W$ is $\sigma$-sub-Gaussian if its Orlicz norm $\|W\|_{\psi_2} := \sup_{k \geq 1} k^{-1/2} \left( \mathbb{E}|W - \mathbb{E}W|^k \right)^{1/k} \leq \sigma$, which is equivalent to the standard definition of sub-Gaussian random variable, see [24]. Furthermore, we use standard Landau's asymptotic notations as follows: for two non-negative sequences $\{a_n\}$ and $\{b_n\}$, let $a_n = O(b_n)$ iff $\limsup_{n \to \infty} a_n/b_n < \infty$, $a_n = \Theta(b_n)$ iff $a_n = O(b_n)$ and $b_n = O(a_n)$, and $a_n = \widetilde{O}(b_n)$ iff for some $a_n = O(b_n)$ up to a poly-log factor of $b_n$.

We assume that the data $\{(X_i, Y_i)\}_{i=1}^n$ are i.i.d samples from $P$. To analyze the asymptotic behavior of the DRCR estimator, we shall impose the following assumptions on the distribution of $X$ and the random variable $\mathcal{E}$ in (1).

**Assumption 1.** *There exists some $\alpha, \gamma > 0$ such that*

$$\mathbb{E} \exp \left( \alpha \|X\|_\infty^\gamma \right) < \infty. \tag{9}$$

**Assumption 2.** *The distribution of $\mathcal{E}$ is $\sigma$-sub-Gaussian for some $\sigma > 0$, symmetric about zero, and has a continuous positive density $p_\mathcal{E}(\cdot)$ in a neighborhood of $0$.*

**Remark 1.** *Assumption 1 allows the study of random variables (such as Weibull random variables) exhibiting heavy tail behavior [9].*

**Remark 2.** *The assumptions on the symmetry and the density, ensure that $0$ is the unique median of $\mathcal{E}$. As is standard in statistical formulations involving absolute error minimization, this assumption is needed to guarantee the consistency of our estimator.*

In the rest of this section, we study the convergence rate of the DRCR estimator $\widehat{f}_{n,\delta_n}$ introduced in Section 2.2. We consider the general question of convergence rate for robustified estimators of the form

$$\widehat{g}_{n,\delta_n}(x) \in \arg\min_{f \in \mathcal{F}_n} \left\{ \sup_{P \in \mathcal{P}(\mathbb{R}^{d+1}): D_c(P,P_n) \leq \delta_n} \mathbb{E}_P \left[ l(Y, f(X)) \right] \right\}. \tag{10}$$

We will show that by a suitable choice of $\delta_n$, the convergence rate of $\widehat{g}_{n,\delta_n}$ to $f_*$ under the empirical $l_1$ loss is of order $\widetilde{O}\left(n^{-1/d}\right)$, where the empirical $l_1$ loss of any two functions $f, g$ is defined as

$$l_1(f, g) := \frac{1}{n} \sum_{i=1}^n |f(X_i) - g(X_i)|.$$

Now we state our main theorem. The proof details are deferred to the supplementary materials (Appendix A).

**Theorem 2.** *If $\|\nabla f_*\|_\infty < \infty$ and $d > 4$, and Assumption 1 and 2 hold, we can pick a $\delta_n$ of order $\Theta(n^{-\frac{2}{d}}(\log n)^{1+\frac{3}{\gamma}})$ so that for any $\widehat{g}_{n,\delta_n}(\cdot)$ defined via (10), there exists some constant $C > 0$ such that*

$$\mathbb{P}\left( l_1(\widehat{g}_{n,\delta_n}, f_*) > C n^{-\frac{1}{d}} (\log n)^{\frac{\gamma+3}{2\gamma}} \right) \to 0 \quad \text{as } n \to \infty. \tag{11}$$

In particular, the DRCR estimator $\widehat{f}_{n,\delta_n}$ defined in (8) also enjoys the rate of $\widetilde{O}(n^{-1/d})$, which is the best known rate so far (compare to [3, 13, 15]). In contrast to prior work, the estimation are not defined in terms of a priori bounds on $\|f_*\|_\infty$ and $\|\nabla f_*\|_\infty$.

# 3 Numerical Experiments

## 3.1 Synthetic datasets

In this section we investigate the performance of our estimator $\widehat{f}_{n,\delta}$, and compare it with the least squares estimator (LSE) of convex regression in [15], as well as the kernel smoothing estimator. We conduct the experiments in the following setting. For each $d$ and $n$, we generate i.i.d. random variables $X_i \in \mathbb{R}^d, i = 1 \ldots n$ such that each coordinate of $X_i$ are i.i.d. from $N(0,1)$, or a standard Student's t-distribution with 10 degrees of freedom. We include this heavy-tailed specification to empirically test the impact of Assumption 1 in our estimator. The results suggest that even if such assumption is violated, our estimator still performs remarkably well.

Let $f_* : \mathbb{R}^d \to \mathbb{R}$ such that

$$f_*(x) = \sum_{i=1}^{d} |x_i|, \quad x = (x_1, \ldots, x_d).$$

We generate $Y_i, i = 1 \ldots d$ by $Y_i = f_*(X_i) + \mathcal{E}_i$, where the noises $\mathcal{E}_i$ are sampled i.i.d. from $N(0, \sigma^2)$.

We construct our DRCR estimator $\widehat{f}_{n,\delta_n}$ by taking $\delta_n = n^{-2/d}$. For the LSE of convex regression, in line with the setting in [3, 15], let $c$ be any numerical constant greater than $\|\nabla f_*\|_\infty$, and we consider the class of functions

$$\mathcal{F}_c := \{f : f \text{ is convex}, \|\nabla f\|_\infty \leq c\}.$$

Let $\widehat{f}_{n,c}^{\text{LS}}$ be the least squares convex regression estimator, namely,

$$\widehat{f}_{n,c}^{\text{LS}} = \arg\min_{f \in \mathcal{F}_c} \left\{ \frac{1}{n} \sum_{i=1}^{n} (Y_i - f(X_i))^2 \right\}.$$

In [3, 15] it is shown that $\widehat{f}_{n,c}^{\text{LS}}$ converges to $f_*$ for any $c > \|\nabla f_*\|_\infty$. Given that $\|\nabla f_*\|_\infty = 1$, we set $c = 10$ or $0.8$, since in practice we typically do not have a tight bound for $\|\nabla f_*\|_\infty$ (we may overestimate/underestimate $\|\nabla f_*\|_\infty$).

Next we construct the kernel regression estimator. Although not required to be convex, the kernel estimator is a good benchmark comparison choice, in the non-parametric setting. For some bandwidth $h_n > 0$, we define the kernel regression estimator $\widehat{k}_{n,h_n}$ by $\widehat{k}_{n,h_n}(x) = \sum_{i=1}^{n} Y_i K(\frac{x-X_i}{h_n}) / \sum_{i=1}^{n} K(\frac{x-X_i}{h_n})$, where $K : \mathbb{R}^d \to \mathbb{R}$ denotes the Gaussian kernel with $K(x) = (2\pi)^{-\frac{d}{2}} e^{-\|x\|^2/2}$. We then choose the best bandwidth $h_n$ via cross validation. To be specific, we pick $h_n = Cn^{-\frac{1}{d+4}}$, and then optimize the choice $C$ via line search. That is, for each $1 \leq j \leq n$, let $\widehat{k}_{n,h_n}^{(-j)}(x) = \sum_{i=1,i\neq j}^{n} Y_i K(\frac{x-X_i}{h_n}) / \sum_{i=1,i\neq j}^{n} K(\frac{x-X_i}{h_n})$ and we select $C$ to be the minimizer of

$$\min_{C \in \{j/100, 1 \leq j \leq 100\}} \sum_{i=1}^{n} \left( Y_i - \widehat{k}_{n,Cn^{-1/(d+4)}}^{(-i)}(X_i) \right)^2.$$

Define the empirical $l_2$ loss of any two functions $f, g$ as

$$l_2(f,g) := \left( \frac{1}{n} \sum_{i=1}^{n} |f(X_i) - g(X_i)|^2 \right)^{\frac{1}{2}}.$$

In the experiments, we set $d = 5$, $n \in \{50, 100, 150, 200, 250, 300, 350\}$ and $\sigma = 0.2$. We compare the performance of $\widehat{f}_{n,\delta_n}, \widehat{f}_{n,0.8}^{\text{LS}}, \widehat{f}_{n,10}^{\text{LS}}$ and $\widehat{k}_{n,h_n}$ under both the empirical $l_1$ and $l_2$ losses. For each choice of $n$ and $d$, we repeat the simulation 100 times and calculate their average.

We first sample i.i.d. $X_i \sim N(0, I_d)$ for the light tail case that satisfying Assumption 1. To compare, we also sample i.i.d. heavy tail random variable $X_i$ such that coordinates of $X_i$ are i.i.d. from the t-distribution with parameter 10. The results of the experiment follow.

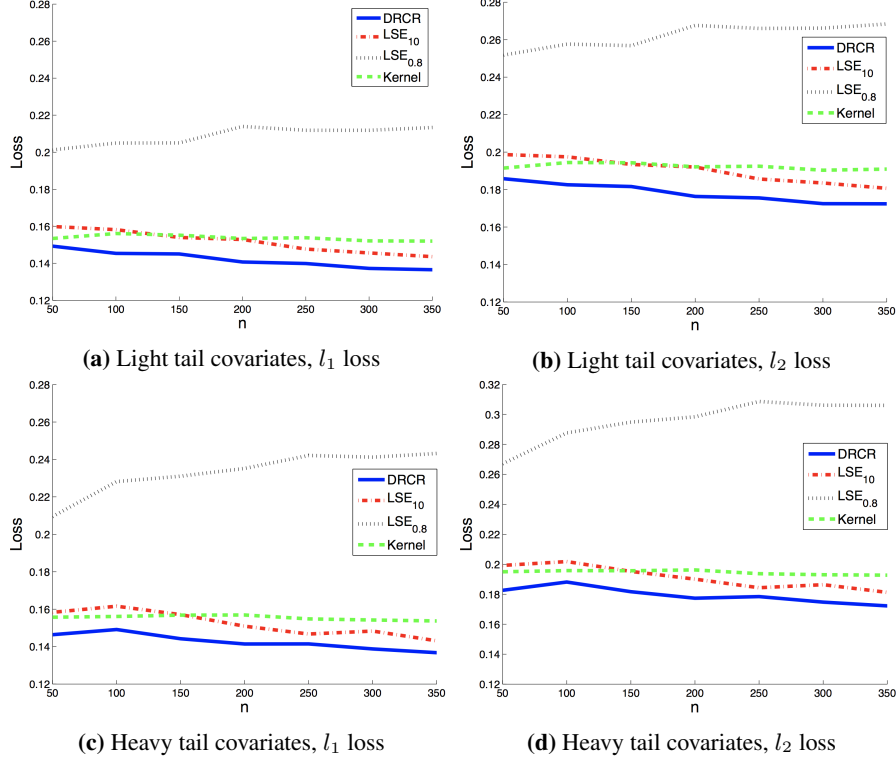

**(a)** Light tail covariates, $l_1$ loss  **(b)** Light tail covariates, $l_2$ loss

**(c)** Heavy tail covariates, $l_1$ loss  **(d)** Heavy tail covariates, $l_2$ loss

**Figure 1:** In the above plots, the blue solid line stands for the estimator $\widehat{f}_{n,\delta}$, the black dotted line stands for $\widehat{f}^{\mathrm{LS}}_{n,0.8}$, the red dash-dot line stands for the estimator $\widehat{f}^{\mathrm{LS}}_{n,10}$, and the green dashed line stands for the kernel estimator $\widehat{k}_{n,h_n}$.

From the Figure 1 in above, we observed that our estimator $\widehat{f}_{n,\delta}$ outperforms $\widehat{f}^{\mathrm{LS}}_{n,0.8}$, $\widehat{f}^{\mathrm{LS}}_{n,10}$ and $\widehat{k}_{n,h_n}$ in both $l_1$ and $l_2$ losses, and the performance of the least squares estimator is highly sensitive to the choice of the constant $c$, the a priori bound on $\|\nabla f_*\|_\infty$. We believe that a key factor in the performance of our estimator is the regularization penalty introduced in the DRCR formulation.

## 3.2 Real dataset

We consider a public dataset from United States Environmental Protection Agency, which was suggested by [17]. The dataset consists of 600 air market data of California in the first quarter of 2019. The response was the amount of heat input with the covariates corresponding to the amounts of emissions of SO2, NOx, CO2 (in tons) and the NOX rate. Empirical evidence suggests that relationship between the response and the log transformation of each individual covariate can be modeled well by a convex fit, so we do the log transformation on covariates and then standardize the data. Since we never know $f^*$ in real data, we can not evaluate our method in the same way as the submitted paper. Instead, we randomly split the dataset into a training set with 400 data and a test set with 200 data, and we implement three different approaches: DRCR, LSE and LR (linear regression). We repeat the experiment 10 times and then compare the average training $l_1$ loss and average test $l_1$ error.

| Method | Training loss | Test error |
|--------|--------------|------------|
| DRCR | **0.1238** | **0.1294** |
| LSE | 0.1485 | 0.1516 |
| LR | 0.1691 | 0.1692 |

We summarize the results in the above table. It is clear that our method outperforms both LSE and LR.

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
