[Supplementary Material · neurips_supplementary.pdf]

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

## Appendix A. Proof of Theorem 2.

In this section we present the full proof of Theorem 2. To begin, we introduce the following lemmas. Their proofs are deferred to Appendix B.

**Lemma 2.** *Under Assumption 1,*

$$\mathbb{P}\left(\sup_{1\leq i\leq n}\|X_i\|_\infty < \frac{1}{2}\left(\log n\right)^{\frac{3}{\gamma}}\right) \to 1,$$

*as $n \to \infty$.*

In the arguments below, we define $\overline{\mathbb{P}}_n$ to be the conditional probability $\mathbb{P}(\cdot|X_1, \cdots, X_n)$, and $\overline{\mathbb{E}}_n$ to be the conditional expectation $\mathbb{E}(\cdot|X_1, \cdots, X_n)$.

**Lemma 3.** *If*

$$\Gamma_0 = \left\{\widehat{g}_{n,\delta_n}(X_i) > \sup_{1\leq i\leq n}|f_*(X_i)| + 1, \ \forall i \in [n]\right\}$$

$$\cup \left\{\widehat{g}_{n,\delta_n}(X_i) < -\sup_{1\leq i\leq n}|f_*(X_i)| - 1, \ \forall i \in [n]\right\}$$

*then*

$$\mathbb{P}(\Gamma_0) \leq 2e^{-2n(\frac{1}{2}-p)^2},$$

*where $p := \mathbb{P}(\mathcal{E}_i \geq 1)$.*

Now we define the set of interest

$$\mathcal{L}_n := \left\{f : f \text{ is convex}, \|\nabla f\|_\infty \leq \log n, \|f\|_\infty \leq 1 + \sup_{\|x\|_\infty \leq (\log n)^{\frac{3}{\gamma}}}|f_*(x)| + (\log n)^{1+\frac{3}{\gamma}}\right\}.$$

By Lemma 2 and Lemma 3, we see that

$$\mathbb{P}\left(\widehat{g}_{n,\delta_n} \in \mathcal{L}_n\right) \to 1. \tag{12}$$

For each function $f \in \mathcal{L}_n$, denoted by

$$Z_n(f) = \frac{1}{n}\sum_{i=1}^n \overline{\mathbb{E}}_n\left(|f_*(X_i) - f(X_i) + \mathcal{E}_i| - |\mathcal{E}_i|\right),$$

and

$$Y_n(f) = \frac{1}{n}\sum_{i=1}^n \left(|f_*(X_i) - f(X_i) + \mathcal{E}_i| - |\mathcal{E}_i|\right) - Z_n(f).$$

We need two basic properties of $Z_n(f)$ and $Y_n(f)$. The proofs can be found in Appendix B.

**Lemma 4.** *For any functions $f, g \in \mathcal{L}_n$ and all $t \geq 0$,*

$$\overline{\mathbb{P}}_n\left(Y_n(f) - Y_n(g) \geq t\right) \vee \overline{\mathbb{P}}_n\left(Y_n(f) - Y_n(g) \leq -t\right)$$

$$\leq \quad \exp\left(-\frac{cnt^2}{\frac{1}{n}\sum_{i=1}^n |f(X_i) - g(X_i)|^2 \wedge (16\sigma^2)}\right).$$

*Where $\sigma$ is the sub-Gaussian parameter of $\mathcal{E}$, and $c$ is some numerical constant (independent of $f, g$ and $n$).*

**Lemma 5.** *There exists a constant $c_0 > 0$, such that for each $f$ with $l_1(f, f_*) > \sigma_n$, we have that*

$$Z_n(f) \geq c_0\sigma_n^2.$$

By the definition of $\widehat{g}_{n,\delta_n}$, we have

$$\delta_n\|\nabla\widehat{g}_{n,\delta_n}\|_\infty + \frac{1}{n}\sum_{i=1}^n l(Y_i, \widehat{g}_{n,\delta_n}(X_i)) \leq \delta_n\|\nabla f_*\|_\infty + \frac{1}{n}\sum_{i=1}^n l(Y_i, f_*(X_i)),$$

which implies
$$Y_n\left(\widehat{g}_{n,\delta_n}\right) + Z_n\left(\widehat{g}_{n,\delta_n}\right) + \delta_n(\|\nabla\widehat{g}_{n,\delta_n}\|_\infty - \|\nabla f_*\|_\infty) \le 0.$$
Together with (12), it suffices to show that
$$\overline{\mathbb{P}}_n\left(\inf_{f\in\mathcal{L}:l_1(f,f_*)>\sigma_n} Y_n\left(f\right) + Z_n\left(f\right) + \delta_n(\|\nabla f\|_\infty - \|\nabla f_*\|_\infty) \le 0\right) \to 0, \quad \text{as } n \to \infty, \quad (13)$$
where $\sigma_n$ is chosen as
$$\sigma_n = \frac{\sqrt{2\delta_n\left(\|\nabla f_*\|_\infty \vee 1\right)}}{c_0}, \tag{14}$$
and $\delta_n$ to be determined later. Given the choice of $\sigma_n$, we may assume $\|\nabla f_*\|_\infty \ge 1$ in the rest of the proof. To carefully bound (13), we apply the following covering lemma.

**Lemma 6** ([12])**.** *Let $\mathcal{C}([a,b]^d, B, L)$ denotes the class of real-valued convex functions defined on $[a,b]^d$ that are uniformly bounded in absolute value by $B$ and uniformly Lipschitz with constant $L$, then*
$$M\left(\mathcal{C}([a,b]^d, B, L), \varepsilon; \rho\right) \le \exp\left(c_1\left(\frac{\varepsilon}{B + L(b-a)}\right)^{-d/2}\right),$$
*where $c_1$ is a constant independent of $a, b, B, L$ and $\varepsilon$.*

Denote by $\rho_n$ the metric such that
$$\rho_n(f,g) := \sup_{\|x\|_\infty \le (\log n)^{\frac{3}{\gamma}}} |f(x) - g(x)|.$$

By Lemma 6, together with the fact that $\sup_{\|x\|_\infty \le (\log n)^{3/\gamma}} \|f_*\|$ is of order $\|\nabla f_*\|_\infty (\log n)^{\frac{3}{\gamma}}$, we have for $n$ large enough, given any $\varepsilon > 0$, there exists an $\varepsilon$-covering $A_\epsilon$ of the set $\mathcal{L}_n$ under metric $\rho_n$, such that
$$|A_\varepsilon| \le \exp\left(c_1\left(\frac{\varepsilon}{1 + \sup|f_*(x)\mathbb{1}(\|x\|_\infty \le (\log n)^{\frac{3}{\gamma}})| + 3(\log n)^{1+3/\gamma}}\right)^{-\frac{d}{2}}\right)$$
$$\le \exp\left(c_1\left(\frac{\varepsilon}{4(\log n)^{1+3/\gamma}}\right)^{-\frac{d}{2}}\right).$$
holds for $n$ is sufficiently large. For each $j \ge 0$, define
$$\varepsilon_j = 2^{-j}\varepsilon_0. \tag{15}$$
where $\varepsilon_0 > 0$ to be determined later. For any $N \ge 1$, we have the following decomposition
$$Y_n(f) = Y_n(f_0) + \sum_{i=0}^{N-1}(Y_n(f_{i+1}) - Y_n(f_i)) + (Y_n(f) - Y_n(f_N))$$
holds for all $f_i \in A_{\varepsilon_i}$ $(0 \le i \le N)$. In particular, we can choose $f_{i+1} \in A_{\varepsilon_{i+1}}$ such that $\rho(f_{i+1}, f) < \varepsilon_{i+1}$ for all $i \ge 1$. By the choice of $\sigma_n$ in (14), together with Lemma 5 as well as the union bound, we conclude that
$$\overline{\mathbb{P}}_n\left(\inf_{f\in\mathcal{L}:l_1(f,f_*)>\sigma_n} Y_n\left(f\right) + Z_n\left(f\right) + \delta_n(\|\nabla f\|_\infty - \|\nabla f_*\|_\infty) \le 0\right)$$
$$\le \overline{\mathbb{P}}_n\left(\inf_{f\in\mathcal{L}:l_1(f,f_*)>\sigma_n} Y_n(f) + \delta_n\|\nabla f_*\|_\infty \le 0\right)$$
$$\le \sum_{f_0\in A_{\epsilon_0}} \overline{\mathbb{P}}_n\left(Y_n(f_0) \le -\frac{\delta_n\|\nabla f_*\|_\infty}{3}\right)$$
$$+ \sum_{j=0}^{N-1}\sum_{\substack{f_j\in A_{\varepsilon_j}, f_{j+1}\in A_{\varepsilon_{j+1}}, \\ \rho(f_j, f_{j+1})<2\varepsilon_j}} \overline{\mathbb{P}}_n\left(Y_n(f_{j+1}) - Y_n(f_j) \le -t_j\right)$$
$$+ \sum_{f_N\in A_{\varepsilon_N}} \overline{\mathbb{P}}_n\left(\inf_{f:\rho(f,f_N)<\varepsilon_N} Y_n(f) - Y_n(f_N) \le -\frac{\delta_n\|\nabla f_*\|_\infty}{3}\right)$$
$$:= I_1 + I_2 + I_3. \tag{16}$$

where $t_j > 0$ will be chosen later so that

$$\sum_{j=0}^{N-1} t_j \leq \frac{\delta_n \|\nabla f_*\|_\infty}{3}. \tag{17}$$

Next we show that (16) goes to zero. Let us begin with a proper choice of $\varepsilon_0$, $N$, $t_j (0 \leq j \leq N-1)$ and $\delta_n$. Let $\varepsilon_0$ satisfy

$$c_1 \left( \frac{\varepsilon_0}{4(\log n)^{1+\frac{3}{\gamma}}} \right)^{-\frac{d}{2}} = \frac{cn \left( \delta_n \|\nabla f_*\|_\infty \right)^2}{288\sigma^2},$$

so that,

$$\varepsilon_0 = 4 \left( \frac{288c_1\sigma^2}{c\|\nabla f_*\|_\infty^2} \right)^{\frac{2}{d}} (\log n)^{1+\frac{3}{\gamma}} \delta_n^{-\frac{4}{d}} n^{-\frac{2}{d}}.$$

Furthermore, we define $t_j$ so that

$$2c_1 \left( \frac{\varepsilon_{j+1}}{4(\log n)^{1+\frac{3}{\gamma}}} \right)^{-\frac{d}{2}} = \frac{cnt_j^2}{8\varepsilon_j^2},$$

that is,

$$t_j = 4c_1^{\frac{1}{2}} c^{-\frac{1}{2}} 8^{\frac{d}{4}} \varepsilon_j^{1-\frac{d}{4}} (\log n)^{\frac{d}{4}(1+\frac{3}{\gamma})} n^{-\frac{1}{2}}.$$

Finally, we set

$$\delta_n = 96 \left( \frac{c_1}{c} \right)^{\frac{2}{d}} (\log n)^{1+\frac{3}{\gamma}} n^{-\frac{2}{d}}.$$

and define

$$N_n = \inf \left\{ N \geq 0 : \varepsilon_0 2^{-N} < \frac{\delta_n \|\nabla f_*\|_\infty}{6} \right\}.$$

Now we are able to bound $I_1$, $I_2$ and $I_3$ in (16) accordingly.

1.*Upper bound for $I_1$*. The choice of $\varepsilon_0$, together with Lemmas 4 and 6, implies that

$$I_1 \leq \exp \left( c_1 \left( \frac{\varepsilon_0}{4(\log n)^{1+\frac{3}{\gamma}}} \right)^{-\frac{d}{2}} - \frac{cn \left( \delta_n \|\nabla f_*\|_\infty \right)^2}{144\sigma^2} \right)$$

$$= \exp \left( -\frac{cn \left( \delta_n \|\nabla f_*\|_\infty \right)^2}{288\sigma^2} \right). \tag{18}$$

2. *Upper bound for $I_3$*. We first check that $N_n > 1$ when $n$ sufficiently large. To see this, note that the definition of $N_n$ implies that

$$\varepsilon_0 > \frac{\delta_n \|\nabla f_*\|_\infty}{6},$$

that is,

$$4 \left( \frac{288c_1\sigma^2}{c\|\nabla f_*\|_\infty^2} \right)^{\frac{2}{d}} (\log n)^{1+\frac{3}{\gamma}} \delta_n^{-\frac{4}{d}} n^{-\frac{2}{d}} > \frac{\delta_n \|\nabla f_*\|_\infty}{6}$$

which is equivalent to

$$\frac{24^{\frac{d}{2}} 288c_1\sigma^2}{c\|\nabla f_*\|^{2+\frac{d}{2}}} n^{\frac{4}{d}} (\log n)^{\frac{(\gamma+3)d}{2\gamma} - (2+\frac{d}{2})(1+\frac{3}{\gamma})} > 1.$$

The above inequality holds trivially for sufficiently large $n$. Note that for any $f$ such that $\rho(f, f_{N_n}) < \varepsilon_{N_n}$, we have

$$|Y_n(f) - Y_n(f_{N_n})| \leq 2\varepsilon_{N_n} < \frac{\delta_n \|\nabla f_*\|_\infty}{3}.$$

Hence

$$\inf_{f:\rho(f,f_{N_n})<\varepsilon_{N_n}} Y_n(f) - Y_n(f_{N_n}) > -\frac{\delta_n \|\nabla f_*\|_\infty}{3}.$$

which simply makes $I_3 = 0$.

3.*Upper bound for $I_2$*. For any $1 \le j \le N_n - 1$, the choice of the $f_j$'s implies that

$$\frac{1}{n} \sum_{i=1}^{n} |f_j(X_i) - f_{j+1}(X_i)|^2 \wedge (4\sigma)^2 \le \frac{1}{n} \sum_{i=1}^{n} |f_j(X_i) - f_{j+1}(X_i)|^2 \le 4\varepsilon_j^2.$$

By the choice of the $t_j$'s, together with Lemmas 4 and 6, we have

$$
\begin{aligned}
I_2 &\le \sum_{j=0}^{N_n-1} \exp\left( 2c_1 \left( \frac{\varepsilon_{j+1}}{4(\log n)^{1+\frac{3}{\gamma}}} \right)^{-\frac{d}{2}} - \frac{cnt_j^2}{4\varepsilon_j^2} \right) \\
&= \sum_{j=1}^{N_n-1} \exp\left( -2c_1 \left( \frac{\varepsilon_{j+1}}{4(\log n)^{1+\frac{3}{\gamma}}} \right)^{-\frac{d}{2}} \right) \\
&= \sum_{j=1}^{N_n-1} \exp\left( -2c_1 \left( \frac{\varepsilon_0}{4(\log n)^{1+\frac{3}{\gamma}}} \right)^{-\frac{d}{2}} 2^{\frac{(j+1)d}{2}} \right) \\
&= \sum_{j=1}^{N_n-1} \exp\left( -\frac{cn\left(\delta_n \|\nabla f_*\|_\infty\right)^2}{144\sigma^2} 2^{\frac{(j+1)d}{2}} \right)
\end{aligned}
\tag{19}
$$

Next, we verify that (17) holds. Note that $t_j = t_0 2^{(\frac{d}{4}-1)j}$ for all $0 \le j \le N_n - 1$. Hence

$$
\begin{aligned}
\sum_{j=0}^{N_n-1} t_j &= t_0 \frac{2^{(\frac{d}{4}-1)N_n} - 1}{2^{\frac{d}{4}-1} - 1} \\
&\le t_0 \frac{2^{(\frac{d}{4}-1)N_n}}{2^{\frac{d}{4}-1} - 1} \\
&= 4 \left( \frac{c_1}{c} \right)^{\frac{1}{2}} 8^{\frac{d}{4}} \left( \varepsilon_0 2^{-N_n} \right)^{1-\frac{d}{4}} (\log n)^{\frac{d}{4}(1+\frac{3}{\gamma})} n^{-\frac{1}{2}}.
\end{aligned}
\tag{20}
$$

By definition of $N_n$ (note that $N_n > 1$), we have $\varepsilon_0 2^{-N_n} > \frac{1}{12} \delta_n \|\nabla f_*\|_\infty$. By substituting this into (20), it suffices to check that

$$4 \left( \frac{c_1}{c} \right)^{\frac{1}{2}} 8^{\frac{d}{4}} \left( \frac{\delta_n \|\nabla f_*\|_\infty}{12} \right)^{1-\frac{d}{4}} a_n^{\frac{d}{4}} (\log n)^{\frac{3d}{4\gamma^*}} n^{-\frac{1}{2}} \le \frac{\delta_n \|\nabla f_*\|_\infty}{3},$$

which is equivalent to

$$\delta_n \|\nabla f_*\|_\infty \ge 96 \left( \frac{c_1}{c} \right)^{\frac{2}{d}} (\log n)^{1+\frac{3}{\gamma}} n^{-\frac{2}{d}}. \tag{21}$$

The above holds because of our choice of $\delta_n$. (Note that we already assume $\|\nabla f_*\|_\infty \ge 1$, without loss of generality).

Finally, we bound the sum of $I_1, I_2$ and $I_3$ in (16). By (18), (19) and the fact that $I_3 = 0$, we have

$$
\begin{aligned}
I_1 + I_2 + I_3 &\le \exp\left( -\frac{cn\left(\delta_n \|\nabla f_*\|_\infty\right)^2}{288\sigma^2} \right) + \sum_{j=0}^{N_n-1} \exp\left( -\frac{cn\left(\delta_n \|\nabla f_*\|_\infty\right)^2}{144\sigma^2} 2^{\frac{(j+1)d}{2}} \right). \\
&\le \sum_{j=0}^{\infty} \exp\left( -\frac{cn\left(\delta_n \|\nabla f_*\|_\infty\right)^2}{288\sigma^2} 2^{\frac{jd}{2}} \right).
\end{aligned}
\tag{22}
$$

Note that for any $t > \log 2$,

$$\sum_{j=0}^{\infty} \exp\left( -t 2^{\frac{jd}{2}} \right) \le \sum_{j=0}^{\infty} \exp\left( -t(j+1) \right) \le 2 \exp\left( -t \right). \tag{23}$$

By our choice of $\delta_n$, we have that

$$\frac{cn\left(\delta_n\|\nabla f_*\|_\infty\right)^2}{288\sigma^2} = 32c^{1-\frac{4}{d}}c_1^{\frac{4}{d}}\sigma^{-2}\|\nabla f_*\|_\infty^2\left(\log n\right)^{2+\frac{6}{\gamma}}n^{1-\frac{4}{d}}. \tag{24}$$

Since $d > 4$, when $n$ is large enough, the above term is certainly greater than $\log 2$. Hence, for $\sigma_n = \frac{\sqrt{2\delta_n\left(\|\nabla f_*\|_\infty \vee 1\right)}}{c_0} = \Theta\left(n^{-\frac{1}{d}}\left(\log n\right)^{1+\frac{3}{\gamma}}\right)$, and (16) is bounded by

$$\overline{\mathbb{P}}_n\left(\inf_{f\in\mathcal{L}:l_1(f,f_*)>\sigma_n}Y_n(f) + \delta_n\|\nabla f_*\|_\infty \leq 0\right)$$

$$\leq \quad 2\exp\left(-32c^{1-\frac{4}{d}}c_1^{\frac{4}{d}}\sigma^{-2}\|\nabla f_*\|_\infty^2\left(\log n\right)^{2+\frac{6}{\gamma}}n^{1-\frac{4}{d}}\right),$$

which goes to zero as $n \to \infty$.

□

## Appendix B. Proofs of Lemmas.

In this section, we prove Lemmas 2, 3, 4 and 5 accordingly.

*Proof of Lemma 2.* Since $X_i$'s are i.i.d, Assumption 1 implies that

$$
\begin{aligned}
\mathbb{P}\left(\sup_{1\leq i\leq n}\|X_i\|_\infty < \frac{1}{2}\left(\log n\right)^{\frac{3}{\gamma}}\right) &= \prod_{i=1}^n \mathbb{P}\left(\|X_i\|_\infty < \frac{1}{2}\left(\log n\right)^{\frac{3}{\gamma}}\right)\\
&= \prod_{i=1}^n\left[1 - \mathbb{P}\left(\|X_i\|_\infty \geq \frac{1}{2}\left(\log n\right)^{\frac{3}{\gamma}}\right)\right]\\
&\geq 1 - n\mathbb{P}\left(\|X\|_\infty \geq \frac{1}{2}\left(\log n\right)^{\frac{3}{\gamma}}\right)\\
&\geq 1 - n\exp\left(-\frac{\alpha}{2^\gamma}\left(\log n\right)^3\right)\mathbb{E}\exp\left(\alpha\|X\|_\infty^\gamma\right).
\end{aligned}
$$

Then for $n \geq \exp\left(2^\gamma/\alpha\right)$ we have

$$
\begin{aligned}
1 - n\exp\left(-\frac{\alpha}{2^\gamma}\left(\log n\right)^3\right)\mathbb{E}\exp\left(\alpha\|X\|_\infty^\gamma\right) &\geq 1 - n\exp\left(-\left(\log n\right)^2\right)\mathbb{E}\exp\left(\alpha\|X\|_\infty^\gamma\right)\\
&\geq 1 - \frac{n}{n^{\log n}}\mathbb{E}\exp\left(\alpha\|X\|_\infty^\gamma\right)\\
&\to 1,
\end{aligned}
$$

which complete the proof. $\qquad\square$

*Proof of Lemma 3.* By the definition of $\widehat{g}_{n,\delta_n}(X_i)$ we see that

$$
\sum_{i=1}^n |Y_i - \widehat{g}_{n,\delta_n}(X_i)| = \min_{a\in\mathbb{R}}\left\{\sum_{i=1}^n |Y_i - \widehat{g}_{n,\delta_n}(X_i) - a|\right\},
$$

which implies

$$
\#\{i : Y_i \geq \widehat{f}_{n,\delta_n}(X_i)\} \geq \frac{n}{2}.
$$

Otherwise, we can shift the $\widehat{g}_{n,\delta_n}(X_i)$ by a constant to obtain a smaller objective value, which contradicts the definition of $\widehat{g}_{n,\delta_n}(X_i)$. As a result,

$$
\begin{aligned}
\mathbb{P}\left(\widehat{g}_{n,\delta_n}(X_i) > \sup_{1\leq i\leq n}|f_*(X_i)| + 1, \ \forall i \in [n]\right) &\leq \mathbb{P}\left(\#\{i : Y_i \geq \sup_{1\leq i\leq n}|f_*(X_i)| + 1\} \geq \frac{n}{2}\right)\\
&\leq \mathbb{P}\left(\sum_{i=1}^n \mathbb{1}_{\{\mathcal{E}_i \geq 1\}} \geq \frac{n}{2}\right).
\end{aligned}
$$

Since $\mathcal{E}_i$'s are i.i.d, we have that the $\mathbb{1}_{\{\mathcal{E}_i\geq 1\}}$'s are i.i.d Bernoulli$(p)$. By the symmetry of $\mathcal{E}$, we see that

$$
p := \mathbb{P}(\mathcal{E}_i \geq 1) < \frac{1}{2},
$$

and hence by the Hoeffding's inequality we have that

$$
\mathbb{P}\left(\sum_{i=1}^n \mathbb{1}_{\{\mathcal{E}_i\geq 1\}} \geq \frac{n}{2}\right) \leq e^{-2n\left(\frac{1}{2}-p\right)^2}.
$$

Using the same argument, we get the same bound for

$$
\mathbb{P}\left(\widehat{g}_{n,\delta_n}(X_i) < -\sup_{1\leq i\leq n}|f_*(X_i)| - 1, \ \forall i \in [n]\right),
$$

which complete the proof. $\qquad\square$

*Proof of Lemma 4.* Define

$$h_{\mathcal{E}}(x) := |x + \mathcal{E}| - |\mathcal{E}| - \mathbb{E}\left(|x + \mathcal{E}| - |\mathcal{E}|\right),$$
$$l_{\mathcal{E}}(x) := |x + \mathcal{E}| - |x|.$$

Now we rewrite $Y_n(f) - Y_n(g)$ by

$$Y_n(f) - Y_n(g) = \frac{1}{n}\sum_{i=1}^{n}\left[h_{\mathcal{E}_i}\left(f_*(X_i) - f(X_i)\right) - h_{\mathcal{E}_i}\left(f_*(X_i) - g(X_i)\right)\right] \tag{25}$$

Note that the summands in (25) are i.i.d, and $\|Y\|_{\psi_2} \leq M$ implies $\log \mathbb{E}\exp(t(Y - \mathbb{E}Y)) = O(t^2 M^2)$ for all $t \geq 0$. It suffices to show that

$$\left\| h_{\mathcal{E}_i}\left(f_*(X_i) - f(X_i)\right) - h_{\mathcal{E}_i}\left(f_*(X_i) - g(X_i)\right) \right\|_{\psi_2} \leq |f(X_i) - g(X_i)| \wedge 4\sigma.$$

Observe that the absolute value of the random variable $|f_*(X_i) - f(X_i) + \varepsilon_i| - |f_*(X_i) - g(X_i) + \varepsilon_i|$ is bounded by $|f(X_i) - g(X_i)|$, so its Orlicz norm is also bounded by $|f(X_i) - g(X_i)|$, which implies

$$\left\| h_{\varepsilon_i}\left(f_*(X_i) - f(X_i)\right) - h_{\varepsilon_i}\left(f_*(X_i) - g(X_i)\right) \right\|_{\psi_2} \leq |f(X_i) - g(X_i)|.$$

On the other hand,

$$h_{\mathcal{E}_i}\left(f_*(X_i) - f(X_i)\right) - h_{\mathcal{E}_i}\left(f_*(X_i) - g(X_i)\right) = l_{\mathcal{E}_i}\left(f_*(X_i) - f(X_i)\right) - l_{\mathcal{E}_i}\left(f_*(X_i) - g(X_i)\right)$$
$$-\overline{\mathbb{E}}_n\left[l_{\mathcal{E}_i}\left(f_*(X_i) - f(X_i)\right) - l_{\mathcal{E}_i}\left(f_*(X_i) - g(X_i)\right)\right].$$

Note that $|l_{\mathcal{E}_i}\left(f_*(X_i) - f(X_i)\right) - l_{\mathcal{E}_i}\left(f_*(X_i) - g(X_i)\right)| \leq 2|\mathcal{E}_i|$ and $\mathbb{E}|Y - \mathbb{E}Y|^k \leq 2^k \mathbb{E}Y^k$ for any random variable $Y$. We therefore have

$$\left\| h_{\mathcal{E}_i}\left(f_*(X_i) - f(X_i)\right) - h_{\mathcal{E}_i}\left(f_*(X_i) - g(X_i)\right) \right\|_{\psi_2}$$
$$= \sup_{k \geq 1} k^{-1/2}\left(\overline{\mathbb{E}}_n|h_{\mathcal{E}_i}\left(f_*(X_i) - f(X_i)\right) - h_{\mathcal{E}_i}\left(f_*(X_i) - g(X_i)\right)|^k\right)^{1/k}$$
$$\leq \sup_{k \geq 1} k^{-1/2}\left(\overline{\mathbb{E}}_n|l_{\mathcal{E}_i}\left(f_*(X_i) - f(X_i)\right) - l_{\mathcal{E}_i}\left(f_*(X_i) - g(X_i)\right)|^k\right)^{1/k}$$
$$\leq \sup_{k \geq 1} k^{-1/2}2\left(\mathbb{E}|2\mathcal{E}|^k\right)^{1/k} \leq 4\sigma.$$

$\square$

*Proof of Lemma 5.* Define $T : \mathbb{R} \to \mathbb{R}$ such that for any $x \in \mathbb{R}$,

$$T(x) := \mathbb{E}|x + \mathcal{E}| - \mathbb{E}|\mathcal{E}|.$$

By basic calculus, $T'(x) = \mathbb{P}(-x \leq \mathcal{E} \leq x)$, and $T''(x) = p_{\mathcal{E}}(x) + p_{\mathcal{E}}(-x) > 0$ holds for $x$ sufficiently small. Hence $T(x)$ is increasing and convex. In particular, we have

$$T'(0) = 0, \quad T''(0) = 2p_{\mathcal{E}}(0).$$

Note that $p_{\mathcal{E}}(x)$ is continuous around zero, then for $x$ sufficiently small, we have $T''(x) = p_{\mathcal{E}}(x) + p_{\mathcal{E}}(-x) > p_{\mathcal{E}}(0)$. Now we pick $c_0 = \frac{1}{2}p_{\mathcal{E}}(0)$. Then, Taylor's expansion yields

$$T(x) = T(0) + T'(0)x + \frac{1}{2}T''(\eta_x)x^2 \geq c_0 x^2$$

where $\eta_x \in (0, x)$ is some real number. Finally, by the monotonicity and convexity of $T$,

$$Z_n(f) = \frac{1}{n}\sum_{i=1}^{n}T\left(|f_*(X_i) - f(X_i)|\right) \geq T\left(\frac{1}{n}\sum_{i=1}^{n}|f_*(X_i) - f(X_i)|\right) \geq T(\sigma_n) \geq c_0\sigma_n^2.$$

$\square$