[Reviews · NeurIPS 2019]

Reviewer 1



While mostly I was able to understand the contributions, and comparison with the related work is reasonably well-written, I think that the organization and the quality of writing could be significantly improved, as summarized below. a. The main statistical result, Theorem 2, is stated in a very concise manner in the main text. For example, for me it is unclear how $\alpha$ in Assumption 1 influences the result (I guess the ``constant'' C actually depends on it, as well as on some other parameters). b. The structure of the paper could be significantly improved. In particular, discussion of the results and motivation for the estimator are somewhat scattered throughout the text, appearing before the estimator is introduced (see, in particular, lines 29-30, 52-55). c. There are numerous typos, see, e.g., lines 6, 42, display under line 45 (missing subscript of $\inf$), display (4), 92, 101, 177 (missed $\nabla$), 179 and mild stylistic error (``touched'' in line 94, ``order $O(\cdot)$'' and ``at most $O(\cdot)$'' throughout, etc.) Regarding the significance of the contributions and mathematical quality of the paper, I see the following issues. a. First of all, Assumption 1 (Weibull-type tails of the design vector) is relatively weak, and the distrubution robustness of the novel estimator is quite limited. It would be interesting (and more useful in practice) to consider finite moment assumption on the design. I expect that using Huber-type M-estimators, or median-of-means, one can obtain statistically optimal rate $O(-2/(d+4))$ in any dimension (see lines 111-113), perhaps even without explicit Wasserstein regularization, and under finite-moment assumption. This would be in line with the recent parametric results (see, e.g., S. Minsker. Geometric median and robust estimation in Banach spaces), as well as with the empirical results obtained in Sec. 3. Algorithmically, the Huber-loss criterion would result in a quadratic program rather than a linear one. b. While the authors claim that their estimator does not depend on the a priori bound on the size of the gradient of the target function (see, e.g., lines 79-80), such dependence indirectly appears in the constraints of their optimization problem. Also, it is not entirely clear why they impose logarithmic scaling of the gradient. On the positive side, I would like to emphasize the technical quality of the paper, in particular, the proof of Theorem 3 which features a somewhat non-standard chaining argument.

Reviewer 2



Update after reading rebuttal: I appreciate the authors' response to my questions. There are still few points that I would like to discuss here: 1- "our approach also allows l be function maps R^2 to R". I do not see any necessity to complicate the loss function. In the end, the authors consider error-based loss functions where the cost function is in the form of L(y - f(x)), and as far as I know, most lost functions in regression models penalize the prediction error e = y - f(x). 2- regarding to infinite dimensional problem: the loss function \ell(f(X), Y) is Lipschitz with respect to the transportation cost. By the Kantorovich-Rubinstein duality theorem, for any 1-Lipschitz loss function g and Q in the Wasserstein set with parameter \delta we have | - | \leq \delta, where the inner product is a short representation for the integral. Now, by [1, Theorem 2], we know that if \delta = \mathcal{O}(n^{-1/d}, then the data generating distribution P is inside the Wasserstein sett with high probability. Therefore, for any Lipschitz function and with high probability we have | - \leq \delta Denoting the optimizer of (3) by f_n and it optimal value g_n, we have |g_n - | \leq \delta * L \| \nabla f \|_\infty. I should emphasis here that the searching over infinite dimensional decision would not change the guarantees. In this paper, the authors consider a different guarantee as follows | - | \leq C \delta I agree with the authors that their analysis is different, but I want to kindly ask them to add a comments about the previous results in the literature, and show the similarities and differences here. One potential benefit of the proposed approach is that the constant in the concentration rate can be found in closed form while [1, Theorem 2] guarantees the existence of a constant C. I would also like to see a comparison between the concentration bound in [5, Theorem 4.3] and the one in your paper. 3- Given the statistical guarantee and new simulation results, I increase my overall evaluation grade. ======================================= 1- Theorem 1: consider a general convex loss function l(y, z): \mathbb{R} \times \mathbb{R} \to \mathbb{R} and a general convex estimator f(x): \mathbb{R^d} \to \mathbb{R}. In the parametric setting, it is known that if f(x) is an affine function then the distributionally robust problem coincides with the regularized empirical loss minimization problem whenever the loss function is convex and Lipschitz [4, Theorem 3.1, Remark 3.5]. The result is significantly generalized in [2, Theorem 1] where it is shown that convexity of the loss function is not required. Theorem 1 in the current submission offers a new regularization whenever the loss function l and the estimator f are convex. To the best of my understanding, the proof of Theorem 1 is similar to the proof in [2, Theorem 1]. The key point here is that any convex function can be represented by the maximum of infinitely many affine functions thanks to its bi-conjugate. Repeating the proof in [2, Theorem 1] for \tilde\beta = \nabla f(x_0) basically gives the proof of Theorem 1 in this paper. Therefore, it is important to illustrate the similarities and differences between these papers. Moreover, similar result to Lemma 1 can be found (or with simple generalization) in [3, 4, 7]. Please cite these papers as well. 2- The setting in this paper with the transportation cost c(\cdot, \cdot) introduced in equation (2) resembles to the robust or adverserial covariate shift problems, where the ambiguity in the distribution is due to the uncertainty in the distribution of the covariate variable x. Consider the followings: the joint probability distribution P(x, y) (under some technical assumptions) can be written as P(x, y) = \int_y P(x|y) P(y). In my opinion, giving infinite weight to perturbations in y is the same as perturbing P(x|y) while P(y) is fixed to P_n(y). Solving the following distributionally robust problem $$ \inf_f \frac{1}{n} \sup_{D(P_i(x|y_i), \delta_{(x_i, y_i)}) \leq \delta} \sum_{i=1}^n \mathbb{E}_{P_i(x|y_i)} l(y_i, f(x)) $$ gives the exact regularized problem in Theorem 1. I believe it would be instructive to relate this problem to the covariate shift setting. 3- Theorem 2: Assumption 1 is the same as the assumption required for \mathcal{E} in [1, Theorem 2]. Therefore, the consistency and the convergence rate of the estimator can be guaranteed using [4, Theorem 3.4 & 3.5] without imposing assumption 2 and restricting the search space \mathcal{F}. Note that the rate can be further sharpened using the result in [6]. While I like the new proof in the paper, I am not convinced that the new result adds any new insights given that the search space for the convex function f is also restricted to \mathcal{F}_n. As far as I understood, this restriction is crucial to obtain the new concentration bound. 4- \mathcal{F}_n: I don't really understand the necessity of the extra constraint \| \nabla d \|_\infty < \log(n) in Section 2.2. It is not conventional to solve a constrained regularized problem where we have a regularization term in both objective and constraint. As far as I understood, the extra constraint is crucial for Theorem 2 and the statistical bounds. However, from the optimization point of view and using Lagrangian theory, we can always move the constraint to the objective and solve a regularized problem. I was wondering if the authors can comment on this restriction. Moreover, it is necessary to have a theorem for optimality of piecewise affine functions here. While I believe the linear program and representation of convex estimator with an n-piece affine function are correct, I think a formal proof is required here. Moreover, the linear program suggests that the worst-case distribution is an n-point discrete distribution. I was wondering if the authors could comment on the structure of the worst-case distribution. 5- The numerical section is too short, and it doesn't offer any meaningful conclusion. I would suggest to show the actual and the theoretical convergence rate as the function of n. [1] Fournier, Nicolas, and Arnaud Guillin. "On the rate of convergence in Wasserstein distance of the empirical measure." Probability Theory and Related Fields 162.3-4 (2015): 707-738. [2] Gao, Rui, Xi Chen, and Anton J. Kleywegt. "Wasserstein distributional robustness and regularization in statistical learning." arXiv preprint arXiv:1712.06050 (2017). [3] Gao, Rui, and Anton J. Kleywegt. "Distributionally robust stochastic optimization with Wasserstein distance." arXiv preprint arXiv:1604.02199 (2016). [4] Mohajerin Esfahani, Peyman, and Daniel Kuhn. "Data-driven distributionally robust optimization using the Wasserstein metric: Performance guarantees and tractable reformulations." Mathematical Programming 171.1-2 (2018): 115-166. [5] Shafieezadeh-Abadeh, Soroosh, Daniel Kuhn, and Peyman Mohajerin Esfahani. "Regularization via mass transportation." arXiv preprint arXiv:1710.10016 (2017). [6] Weed, Jonathan, and Francis Bach. "Sharp asymptotic and finite-sample rates of convergence of empirical measures in Wasserstein distance." arXiv preprint arXiv:1707.00087 (2017). [7] Zhao, Chaoyue, and Yongpei Guan. "Data-driven risk-averse stochastic optimization with Wasserstein metric." Operations Research Letters 46.2 (2018): 262-267.

Reviewer 3



The paper is well written and is easy to follow. The conducted analysis is also sound.

[Author Response · NeurIPS 2019]

We are grateful to the reviewers for their thoughtful comments which have improved the work. Below are responses to
each and every point of the reviewers (where we refer to the same references in the reviewer 2's comments):

**Reviewer 1**: We thank the reviewer for the positive comments. **1a.** The dependence of the constant $C$ in Theorem 2
on the parameters (such as $\alpha$) is complicated due to the delicate nature of our chaining method, so we did not write
down the constant explicitly. We will carefully address which parameters that the constant $C$ depends on. **1b.** We
appreciate the reviewer's excellent suggestions on the writing structure, we will rewrite lines 29-30 and move lines
52-55 (the motivation of introducing distributional robustness) to follow the formal introduction of our model. **1c.**
We thank the reviewer for pointing out the typos in our paper. The $\inf$ in line 45 refers to the infimum of a set, so
we don't have to add a subscript for it. We will follow the reviewer's advice to correct the other typos. **2a.** Although
the Weibull-type condition is already a significant generalization of related assumptions which are standard in convex
regression formulations, we agree that weakening to finite moments assumption is a very interesting direction to explore
in future work. We also thank the reviewer for bringing up the ideas of robust statistics. In our problem, we care about
the shape of the estimator, it is not clear how estimators such as Huber-type M-estimators can be modified to preserve
the convex shape, but this is also an interesting direction to explore. **2b.** Please see our response to **4** of Reviewer 2.

**Reviewer 2**: We thank the reviewer for the insightful comments. **1.** We appreciate the reviewer for bringing [2] to us,
and we agree that it is important to illustrate the similarities and differences between [2, Theorem 1] and our result. The
main difference is, the loss function in [2, Theorem 1] is required to be a composition of a Lipschitz function $l : \mathbb{R} \to \mathbb{R}$
and a linear function of $x, y$, due to the non-linear nature of our convex regression model, this result does not apply to our
case directly. Besides, our approach also allows $l$ be function maps $\mathbb{R}^2$ to $\mathbb{R}$. However, we agree that the proof of the two
results share a similar idea: we apply the dual representation of the standard Wasserstein-based distributionally robust
optimization problem, and then lower bound the dual parameter $\lambda$ using the structure of the loss function $l$. Moreover,
we will follow the reviewer's advice to cite other results similar to Lemma 1. **2.** The new interpretation of equation (2)
that relates our problem to covariate shifting is very interesting, we will add this point to our submission. **3.** We thank
the reviewer for giving us the chance to illustrate our contribution beyond the results in [4]. First of all, their results only
focus on problems with **finite** dimensional decision space (or feasible set $\mathbb{X}$, in the words of [4]), while in our case the
decision space $\mathcal{F}_n$ is clearly **infinite** dimensional. Secondly, to the best of our understanding, the concentration result
([4, Theorem 3.4], or [1, 6]) allows us to choose a proper $\delta_n$ such that the Wasserstein ball centered at $P_n$ (the empirical
measure) with radius $\delta_n$ covers the true underlying probability measure $P$ (see [4, Theorem 3.5]) with high probability,
and as a consequence we can appropriately choose $\delta_n$ to ensure the **consistency** of the estimator of the optimal decision
variable (see [4, Theorem 3.6(2)]). However, it is non-trivial to identify the convergence rate of the estimator. For
example, for finite dimensional decisions, such convergence rates should match the canonical rate $O(n^{-1/2})$. The choice
of $\delta_n$ suggested in [4] does not provide the canonical rate, so it doesn't seem that the results in [4] are directly applicable
to recover convergence rates for estimators; not even in the finite dimensional case which is the environment of [4], let
alone the infinite dimensional case, which is the setting of our paper. **4.** We thank the reviewer for raising this issue, and
we are agree with the reviewer's opinion from the optimization point of view. However, there is a technical statistical
reason behind our formulation. On the one hand, the penalty term $\delta_n \|\nabla f\|_\infty$ captures the impact of the uncertainty
set $\{P : D(P, P_n) \leq \delta_n\}$. On the other hand, the $\log n$ constraint introduced in $\mathcal{F}_n$ is related to a compactification
argument applied to the decision space. This issue is particularly important in the current setting of infinite dimensional
decisions. The current formulation provides one possible tradeoff of these two effects that guarantee the estimator $\hat{f}_n$
converges to $f_*$ with order $\widetilde{O}(n^{-1/d})$. Furthermore, the $\log n$ constraint already relaxes the typical assumption that
$\|\nabla f\|_\infty < C$ (in which $C$ needs to be known apriori). Finally, we will provide a formal proof of the optimality of
piecewise affine functions. **5.** We appreciate the reviewer's suggestions of adding experiments on real world data. We
consider a public dataset from United States Environmental Protection Agency, which was suggested by [R. Mazumder,
A. Choudhury, G. Iyengar, B. Sen, A Computational Framework for Multivariate Convex Regression and its Variants].
The dataset consists of 600 air market data of California in the first quarter
of 2019. The response was the amount of heat input with the covariates
corresponding to the amounts of emissions of SO2, NOx, CO2 (in tons) and the

| Method | Training loss | Test error |
|--------|---------------|------------|
| DRCR   | **0.1238**    | **0.1294** |
| LSE    | 0.1485        | 0.1516     |
| LR     | 0.1691        | 0.1692     |

NOX rate. Empirical evidence suggests that relationship between the response
and the log transformation of each individual covariate can be modeled well by
a convex fit, so we do the log transformation on covariates and then standardize
the data. Since we never know $f^*$ in real data, we can not evaluate our
method in the same way as the submitted paper. Instead, we randomly split
the dataset into a training set with 400 data and a test set with 200 data, and
we implement three different approaches: DRCR (our estimator), LSE (standard convex regression estimator) and LR
(linear regression). We repeat the experiment 10 times and then compare the average training $l_1$ loss and average test $l_1$
error. We summarize the results in the table on the right, it is clear that our method outperforms both LSE and LR.

**Reviewer 3**: We thank the reviewer for the positive comments.

[Meta-Review · NeurIPS 2019]

The reviewers are quite positive about the contributions, novelty and significance of this submission. They were also globally convinced by the clarifications brought forth in the rebuttal.